# Diverse Sensory Repertoire of Paralogous Chemoreceptors Tlp2, Tlp3, and Tlp4 in *Campylobacter jejuni*

Taha,[a] Bassam A. Elgamoudi,[a] Ekaterina P. Andrianova,[b] Thomas Haselhorst,[a] Christopher J. Day,[a] Lauren E. Hartley-Tassell,[a] Rebecca M. King,[a] Tahria Najnin,[a] Igor B. Zhulin,[b] Victoria Korolik[a,c]

[a]Institute for Glycomics, Griffith University, Gold Coast, Queensland, Australia
[b]Department of Microbiology and Translational Data Analytics Institute, The Ohio State University, Columbus, Ohio, USA
[c]School of Pharmacy and Medical Science, Griffith University, Gold Coast, Queensland, Australia

Taha and Bassam A. Elgamoudi contributed equally to this article. Author order was determined by the corresponding author and does not reflect the level of contribution.

**ABSTRACT** *Campylobacter jejuni* responds to extracellular stimuli via transducer-like chemoreceptors (Tlps). Here, we describe receptor-ligand interactions of a unique paralogue family of dCache_1 (double Calcium channels and chemotaxis) chemoreceptors: Tlp2, Tlp3, and Tlp4. Phylogenetic analysis revealed that Tlp2, Tlp3, and Tlp4 receptors may have arisen through domain duplications, followed by a divergent evolutionary drift, with Tlp3 emerging more recently, and unexpectedly, responded to glycans, as well as multiple organic and amino acids with overlapping specificities. All three Tlps interacted with five monosaccharides and complex glycans, including Lewis's antigens, P antigens, and fucosyl GM1 ganglioside, indicating a potential role in host-pathogen interactions. Analysis of chemotactic motility of single, double, and triple mutants indicated that these chemoreceptors are likely to work together to balance responses to attractants and repellents to modulate chemotaxis in *C. jejuni*. Molecular docking experiments, in combination with saturation transfer difference nuclear magnetic resonance spectroscopy and competition surface plasmon resonance analysis, illustrated that the ligand-binding domain of Tlp3 possess one major binding pocket with two overlapping, but distinct binding sites able to interact with multiple ligands. A diverse sensory repertoire could provide *C. jejuni* with the ability to modulate responses to attractant and repellent signals and allow for adaptation in host-pathogen interactions.

**IMPORTANCE** *Campylobacter jejuni* responds to extracellular stimuli via transducer-like chemoreceptors (Tlps). This remarkable sensory perception mechanism allows bacteria to sense environmental changes and avoid unfavorable conditions or to maneuver toward nutrient sources and host cells. Here, we describe receptor-ligand interactions of a unique paralogue family of chemoreceptors, Tlp2, Tlp3, and Tlp4, that may have arisen through domain duplications, followed by a divergent evolutionary drift, with Tlp3 emerging more recently. Unlike previous reports of ligands interacting with sensory proteins, Tlp2, Tlp3, and Tlp4 responded to many types of chemical compounds, including simple and complex sugars such as those present on human blood group antigens and gangliosides, indicating a potential role in host-pathogen interactions. Diverse sensory repertoire could provide *C. jejuni* with the ability to modulate responses to attractant and repellent signals and allow for adaptation in host-pathogen interactions.

**KEYWORDS** *Campylobacter jejuni*, chemotaxis, ligand discovery, chemoreceptor

Address correspondence to Victoria Korolik, v.korolik@griffith.edu.au.

The authors declare no conflict of interest.

*C*ampylobacter jejuni is one of the most prevalent bacterial causes of gastroenteritis in humans and is estimated to affect 10% of the world's population every year (1, 2). *C. jejuni* can naturally colonize the gastrointestinal tracts of birds and animals, and consumption of undercooked meat and poultry or cross-contaminated food and water has

10.1128/spectrum.03646-22 **1**

been linked to campylobacteriosis in humans (3–5). While *C. jejuni* colonizes the intestinal mucosa, not much is known regarding how *C. jejuni* targets the mucus membranes of host animals and humans.

It is understood that chemotaxis-directed motility and swimming velocity, together with the corkscrew morphology of *C. jejuni*, are important for both host colonization and pathogenicity (6). For chemotaxis events to occur, special receptors or chemoreceptors, also known as methyl-accepting chemotaxis proteins (MCPs) or transducer-like proteins (Tlps), are required to detect the extracellular stimuli. In response, the ligand-binding domain (LBD) of the chemoreceptors initiates signal transduction cascade that propagates the signals through to the cytoplasm and finally modulates the rotation of the flagellar motor proteins, allowing the bacteria to adjust the direction of its movement (7). Several *C. jejuni* chemoreceptors that sense external ligands have been characterized to date: Tlp1 as the ubiquitous receptor for aspartate (7–9), Tlp3 and Tlp10 as multi-ligand-binding receptors (10, 11), Tlp7 as a sensor for formic acid (12), and Tlp11 as a galactose chemoreceptor (13). Among these, the crystal structures of Tlp1 and Tlp3 have been solved, and a number of conserved residues were determined as essential for binding (14, 15).

Chemoreceptors, Tlp1, Tlp2, Tlp3, Tlp4, and Tlp11, belong to a superfamily of double Calcium channels and chemotaxis (dCache_1) domain (11, 16). Although the Cache domains are the most commonly occurring extracellular sensor modules in bacteria, receptors containing the Cache motif are still extremely diverse in both sequence and structure (17, 18). This diversity allows these proteins to recognize and bind a wide range of ligands suggesting a role in efficient bacterial interactions within complex niches. Tlp2, Tlp3, and Tlp4 represent a unique group of paralogue receptors within *C. jejuni* spp. They possess identical signaling domains; however, their periplasmic sensory domains share low sequence similarities (11). Multi-ligand-binding properties of Tlp2 and Tlp3 have been previously reported (11, 19), e.g., Tlp2 showed affinity toward aspartate and pyruvate, while Tlp3 interacted predominantly with amino acids and organic acids. Tlp4, on the other hand, has only been reported to sense bile and deoxycholate (20).

To the best of our knowledge, this is a first report that the Tlp2, Tlp3, and Tlp4 full sensory range includes not only a range of organic/amino acids but also carbohydrates and even more complex sugars (glycans). In fact, this is first report showing, that dCache_1 domain containing proteins can sense and bind monosaccharides and complex glycans. In addition, we demonstrate that Tlp2, Tlp3, and Tlp4 can work in unison to modulate chemotaxis in *C. jejuni*. Our investigation of the evolutionary history of this distinct paralogue family revealed that Tlp2, Tlp3, and Tlp4 receptors may have arisen through domain duplications. We also show that the LBD of Tlp3 contains one major binding pocket with two distinct ligand-specific binding sites that can interact with multiple ligands simultaneously.

## RESULTS

**Phylogenetic relationships of Tlp2, Tlp3, and Tlp4.** Full-length sequence comparison showed that Tlp2, Tlp3, and Tlp4 proteins are >60% identical with respect to each other. The averaged sequence identity was around 30% in the dCache_1 sensory region, but their signaling domains are identical (see Fig. S1 in the supplemental material).

To gain further insight into Tlp2,3,4 chemoreceptors, we analyzed the evolutionary history of the dCache_1-containing chemoreceptors using a phylogenetic approach. A representative set of genomes from *Campylobacterota* phylum (GTDB taxonomy [21]) and all dCache_1-containing chemoreceptors from this set from the MiST3 database (22) (see Table S1) was aligned and used to construct a maximum-likelihood phylogenetic tree using full-length amino acid sequences (see Fig. S2). Considering different rates of evolutionary changes observed between sensory and signaling domains (23), we inferred separate phylogenetic reconstructions for sensory and signaling parts of the chemoreceptors (see Fig. S3 and S4). The phylogenetic tree inferred from signaling

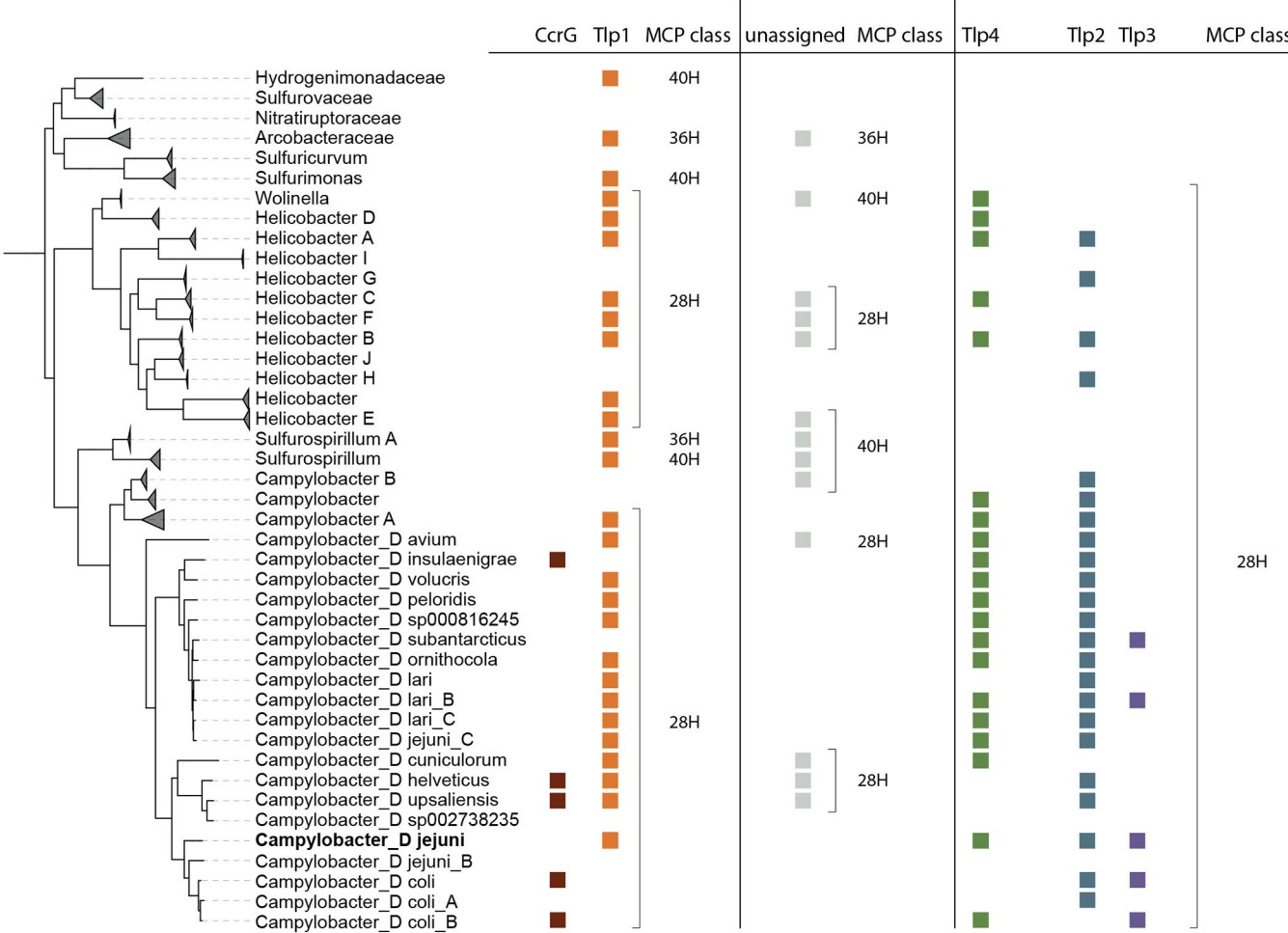

**FIG 1** Phyletic distribution of dCache_1 domains from *Campylobacterota* chemoreceptors. The taxonomy tree, based on 120 protein sequences, was retrieved from AnnoTree (24). *C. jejuni* 11168 belongs to *Campylobacter_D jejuni* and is indicated in boldface. Tlp2,3,4 were assigned to the 28H MCP based on 28 helical heptads in the signaling domain by using hidden Markov models (23). Tlp1 homologs were assigned based on the sequence similarity of the LBD (see Fig. S3); differences in the MCP signaling domain heptad class might be due either domain swap or insertions/deletions.

domains of chemoreceptors mirrored, to a great extent, the most recent genome tree of *Campylobacterota* based on 120 conserved proteins (24) (see Fig. S4). In contrast, a tree built on sensory domains had a different topology, forming two distinct major clusters (see Fig. S3): Tlp2,3,4-like and Tlp1-like, which is in agreement with previous data (13). The Tlp2,3,4-like cluster further formed two clades: one containing Tlp2 and Tlp3 and another containing Tlp4.

In addition, multiple gene duplication events throughout *Campylobacteraceae* family were detected. For instance, *Campylobacter peloridis* LMG 23910 possesses six *tlp2,4*-like chemoreceptors, of which, four dCache domains were closely related to the *tlp4* and the remaining two to *tlp2* (see Table S1 and Fig. S3). The number of chemoreceptors also varies across *C. jejuni* strains (25). While *C. jejuni* 11168-O possesses all three *tlp*'s, there are strains lacking one, two, or all three chemoreceptors. The most common mechanism for the protein loss appears to be gene conversion to a pseudogene (see Table S2). Phylogenetic profiling using 140 *C. jejuni* strains revealed that these chemoreceptors are subject not only to pseudogenization but also to gene duplication (see Table S2). For example, *C. jejuni* strains S3 and NCTC 12660 possess two Tlp3 homologues (see Table S2). Moreover, distribution of these Tlp3 homologues across different branches on the phylogenetic tree indicated that these are independent events (Fig. 1; see also Fig. S2). The *C. jejuni* NCTC 12660 Tlp3 homologue appeared as a result of the recent *tlp3* gene duplication, and it is present only in a few *C. jejuni* strains (Fig. 1).

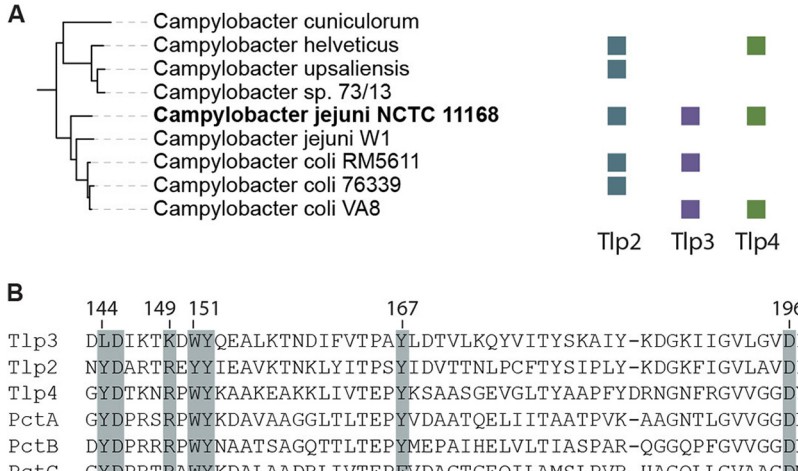

**FIG 2** (A) Phylogenetic distribution of Tlp2, Tlp3, and Tlp4 homologues among *Campylobacter_D* genus. (B) Fragment of dCache_1 sequence alignment of Tlp2, Tlp3, and Tlp4 from *C. jejuni* 11168 and PctA, PctB, and PctC from *P. aeruginosa*. Conserved amino acid recognition motif is highlighted in gray. Amino acid numeration is based on the *tlp3* gene sequence.

**Tlp2 and Tlp4 arose from a common ancestor.** Our results show that orthologues of *tlp2* and *tlp4* genes are present only in a few species of the *Campylobacter_D* genus (Fig. 2A; see also Fig. S2). However, dCache_1 domains that cluster with both Tlp2 and Tlp4 are present in many *Helicobacteraceae* species (Fig. 1; see also Fig. S3), suggesting that duplication of the *tlp2* and *tlp4* ancestor happened before the divergence of *Campylobacteraceae* and *Helicobacteraceae*. After this event, it appears that each duplicate underwent substantial changes in the ligand-binding domain, whereas the cytoplasmic signaling part of chemoreceptors evolved at different rates optimally adapting to chemosensory systems of each taxonomic lineage. Phylogenetic reconstruction showed that Tlp3 originated more recently through *tlp2* gene duplication prior to *C. jejuni* and *C. coli* divergence, which is further supported by the fact that *tlp3* is encoded only in the genomes of these species.

**Presence of a binding motif.** The comparison of the LBDs of all three chemoreceptors with each other and with three amino acid-sensing dCache_1 domains from *Pseudomonas aeruginosa* PAO1: PctA, PctB, and PctC. confirmed the presence of the amino acid recognition motif (AA_motif) reported by Gavira et al. (26, 27) in all three Tlp2,3,4 (Fig. 2B). While the Tlp4 motif was identical to that in PctA and PctB of *P. aeruginosa* PAO1, Trp151 (based on *tlp3* sequence) in the Tlp2 AA_motif was changed to tyrosine. Trp151 was previously shown to be critical for establishing hydrogen bond with carboxyl group of the bound amino acid (26). Of the six identified *tlp2* orthologues, only one recognition motif from *C. helveticus* was identical to the motif from *C. jejuni* 11168-O Tlp2 (see Fig. S3). Similarly, the motif varied in eight *tlp4* orthologues. For example, tyrosine and arginine, which are essential for the hydrogen bond formation with the carboxyl and amino groups, are changed for phenylalanine and glutamine in *C. jejuni* 00-1597. However, aside from such variations, remarkably, all members of the Tlp2,3,4-like cluster possess the amino acid recognition motif (see Fig. S3), which supports our conclusion regarding the common origin of these chemoreceptors.

**Tlp2, Tlp3, and Tlp4 bind to a range of ligands, including glycans and amino acids.** Glycan/small molecule arrays revealed that Tlp2 and Tlp4 bound multiple amino acids (serine, methionine, asparagine, lysine, cysteine and aspartate) and purine, where lysine, purine and aspartate, also bind specifically to Tlp3 (Table 1 and Fig. 3). Considering that Tlp11, a dCache_1 receptor from *C. jejuni* strain 520 (13), and Tlp10 from *C. jejuni* strain 11168-O are able to bind galactose (14), it was decided to assess the ability of Tlp2, Tlp3 (not previously assessed for glycan interactions), and Tlp4 to bind glycans. All three Tlps interacted with the monosaccharides fucose, glucose, galactose, mannose, and sialic acid,

**TABLE 1** Summary of Tlp3, Tlp2, and Tlp4 ligand-binding specificities[a]

| Ligand | Tlp3 | | Tlp2 | | Tlp4 | |
|---|---|---|---|---|---|---|
| | Affinity ($\mu$M) | CR | Affinity ($\mu$M) | CR | Affinity ($\mu$M) | CR |
| Serine | NB | – | 6.9 ± 2.4 | A | 14.3 ± 3.2 | A |
| Methionine | NB | – | 1.9 ± 1.2 | A | 6.0 ± 4.7 | A |
| Asparagine | NB | – | 12.8 ± 5.1 | A | 45.1 ± 6.1 | A |
| Cysteine | NB | – | 10.8 ± 3.3 | NR | 5.7 ± 5.8 | A |
| Lysine | 4.3 ± 1.6 | R | 18.6 ± 1.5 | R | 19.7 ± 3.9 | A |
| Purine | 37.2 ± 17.4 | A | 5.4 ± 1.1 | A | 1.2 ± 0.7 | A |
| Aspartate | 152.3 ± 5.7 | A | 95.4 ± 0.9 | A | 75.3 ± 8.1 | A |
| Malic acid | 17.9 ± 4.2 | A | >10,000 | NR | >10,000 | NR |
| Arginine | 35.3 ± 2.7 | R | NB | – | NB | – |
| Isoleucine | 16.4 ± 1.7 | A | NB | – | NB | – |
| Glucosamine | 7.1 ± 2.9 | R | NB | – | NB | – |
| Thiamine | 41.4 ± 5.9 | R | NB | – | NB | – |
| Glucose | 58.2 ± 9.7 | A | 18.1 ± 2.2 | A | 18.3 ± 1.1 | A |
| Fucose | 25.2 ± 6.5 | A | 15.5 ± 0.9 | A | 16.2 ± 2.2 | A |
| Galactose | 45.1 ± 1.3 | A | 17.2 ± 1.3 | A | 18.3 ± 2.1 | A |
| Mannose | 21.2 ± 2.4 | R | 12.9 ± 1.8 | A | 7.8 ± 1.0 | A |
| Sialic acid | 9.0 ± 0.2 | R | 1.7 ± 0.2 | A | 0.8 ± 0.1 | A |

[a]NB, no binding. The chemoresponse (CR) is indicated as follows: A, attractants; R, repellents, NR, no response; or –, not present. Affinities are expressed as means ± the standard deviations.

and complex glycans with these terminal sugar moieties, such as blood group antigens, Lewis antigens, P antigens, and fucosyl GM1 ganglioside (Table 2; for a full list, see Table S3). Binding to multiple glycans has not been previously reported for any dCache_1 sensory proteins.

SPR analysis showed that the highest affinities, $K_D$ (i.e., the equilibrium dissociation constant; see Fig. S5), for Tlp2 and Tlp4 were detected for interactions with serine, methionine, purine, and sialic acid (Table 1 and Fig. 4; see also Fig. S5). However, strong binding affinities ($K_D < 50$ $\mu$M; Table 1) were also observed for the other amino acids and monosaccharides. The binding affinities of Tlp3 to its previously identified ligands were reassessed using SPR (Cytiva Biacore S200) and were in excellent agreement with the previous report (obtained using Cytiva Biacore T200) (11).

**Tlp3 exhibits one major binding pocket with two distinct ligand-specific binding sites.** In order to understand the binding interactions between Tlp3[LBD] and its ligands, a blind molecular docking modeling was performed using the 'YASARA Structure' molecular modelling package (28–30). Five potential binding clusters were identified which accommodated all the binding ligands (Fig. 3; see also Table S4). Cluster A (ASN 116; distal dCache _1 domain) was identified as a major ligand-binding pocket, and cluster D (PHE 216; proximal dCache _1 domain) was identified as a potential secondary binding pocket (Fig. 3). The amino acid residues (Tyr118, Val126, Lys149, Trp151, Val171, Leu128, Leu144, Asp169, Thr170, and Asp196) that were previously shown to contribute directly to ligand binding to the distal binding domain of Tlp3[LBD] (27) were all identified in cluster A (see Table S4).

To further investigate the Tlp3[LBD] ligand-binding interaction, molecular dynamics simulations between two molecules, a monosaccharide (glucose) and an amino acid (arginine), were conducted (see Videos S1 and S2 in the supplemental material). The modeling showed that both ligands can bind to the same binding pocket (cluster A) (Fig. 5). Furthermore, in the presence of both ligands, arginine bound to a secondary binding site which was adjacent to the binding site for glucose (Fig. 5; see also Table S4). Tlp3[LBD]-arginine interactions were further characterized using saturation transfer difference (STD) nuclear magnetic resonance (NMR) spectroscopy (Fig. 3C). STD NMR analysis of arginine ($K_D \sim 38.6$ $\mu$M) to Tlp3 revealed that the ethylene protons (H3/H3′) adjacent to the carboxylic acid received more saturation from the protein than the ethylene protons (H5/H5′) closer to guanidine group (31, 32). These data were comparable to the previously reported results from the STD NMR experiments of the chemoattractants and chemorepellents in complex with Tlp3[LBD] (22).

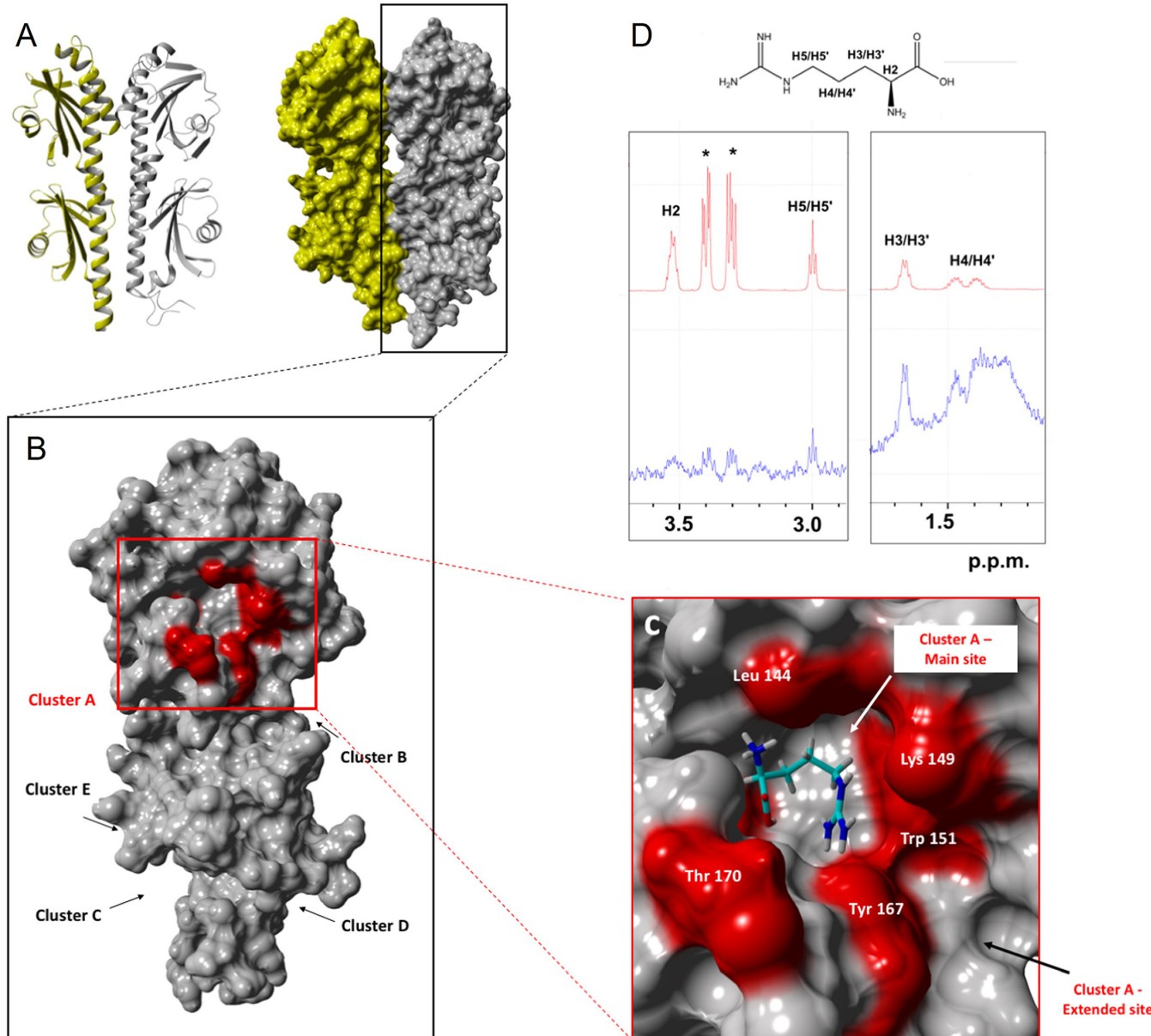

**FIG 3** Structural analysis of arginine binding to Tlp3 using molecular docking experiments and saturation transfer difference (STD) NMR experiments. (A) Ribbon structure of dimeric Tlp3 (PDB 4XMR) (15) (left) and surface structure (right). (B) A monomeric Tlp3 structure was used for an unbiased global docking experiments using glucose and arginine as ligands with all clusters A to D shown, with cluster A being the predominant high-occupancy cluster for both ligands. (C) Zoomed view of cluster A that exhibits a main and an extended binding site with arginine bound to the main binding site. (D) STD (blue) and ¹H NMR spectrum of arginine in complex with Tlp3 acquired at 283K and 600.13 MHz. Strong saturation transfer could be detected for H3/H3′ and H5/H5′ with H4/H4′ overlapped with protein background. The epitope is consistent with the docked structure shown in panel C.

**SPR competition assays confirm one binding pocket with two distinct ligand-binding sites for Tlp3[LBD].** We used competition SPR analysis to determine whether the ligands interact with a single or multiple binding sites of the sensory domain of the Tlp3, Tlp3[LBD]. The ligand-binding site status can be classified as either shared, independent, flexible, or preferential shared sites (10, 33). (i) When two ligands compete for a single site, they are considered to share the binding site. (ii) A preferential shared site is where ligands compete for the same binding site, but one of the ligands binds preferentially. (iii) A flexible binding site is where the ligand binds to a specific site and induces a change in binding site flexibility, allowing another ligand to bind to the second site with lower affinity. (iv) An independent site is where the ligands bind to their own sites. When two ligands bind independently to different sites, the SPR signal is

**TABLE 2** Examples of glycan structures recognized by Tlp2,3,4[a]

| Saccharide category and Tlp2 | Tlp3 | Tlp4 |
|---|---|---|
| **Disaccharides** | | |
| GlcAβ1-3Galβ | | Galα1-3GalNAcα |
| Galβ1-4GlcNAcβ | Galα1-3GalNAcα | Galβ1-6GlcNAc |
| Galβ1-4Gal | | Galβ1-6Gal |
| Fucα1-2Galβ | | Fucα1-3GlcNAcβ |
| Manβ1-4GlcNAcβ | Manβ1-4GlcNAcβ | Manα1-2Manβ |
| Manα1-2Manβ | | |
| **Trisaccharides and tetrasaccharides** | | |
| (Glcα1-4)₃β | (Glcα1-4)₃β | (Glcα1-4)₃β |
| Galα1-3(Neu5Acα2-6)GalNAcα | Galβ1-4GlcNAcβ1-6GalNAcα | Galα1-3(Neu5Acα2-6)GalNAcα |
| Fucα1-3(Neu5Acα2-3Galβ1-4)GlcNAcβ (Sia-Leˣ) | Fucα1-2(Galα1-3)Galβ1-3GlcNAcβ (B [type 1]) | Fucα1-2Galβ1-3(Fucα1-4)GlcNAcβ- (Leᵇ) |
| Fucα1-2(Galα1-3)Galβ1-3GlcNAcβ (B [type 1]) | Neu5Gcα2-3Galβ1-3GlcNAcβ- (3′ Sia-Leᶜ) | Fucα1-2(Galα1-3)Galβ1-3GlcNAcβ (B [type 1]) |
| **Complex glycans** | | |
| Galβ1-4(Fucα1-3)GlcNAcβ1-6(Fucα1-2Galβ1-3GlcNAcβ1-3)Galβ1-4Glc | | Galβ1-4(Fucα1-3)GlcNAcβ1-6(Fucα1-2Galβ1-3GlcNAcβ1-3)Galβ1-4Glc |
| Galβ1-3GlcNAcβ1-3Galβ1-4(Fucα1-3)GlcNAcβ1-3Galβ1-4Glc | | Galβ1-3GlcNAcβ1-3Galβ1-4(Fucα1-3)GlcNAcβ1-3Galβ1-4Glc |
| | | Galβ1-4GlcNAcβ1-2(Galβ1-4GlcNAcβ1-4)Manα1-3(Galβ1-4GlcNAcβ1-2 |
| | | (Galβ1-4GlcNAcβ1-6)Manα1-6Man)β1-4GlcNAcβ1-4GlcNAc |
| | Neu5Acα2-3Galβ1-4GlcNAcβ1-3Galβ1-4GlcNAcβ | |
| Neu5Acα2-6Galβ1-4GlcNAcβ1-2Manα1-3(Neu5Acα2-6Galβ1-4GlcNAcβ1-2Manα1-6)Manβ1-4GlcNAcβ1-4(Fucα1-6)GlcNAc | | |

[a]Fuc, fucose; Man, mannose; Gal, galactose; GalNAc, 2′-N-acetylgalactosamine; Glc, glucose; GlcNAc, 2′-N-acetylglucosamine; Neu5Acα2, sialylated; GlcA, D-glucuronic acid; sp3/4, the linker tethering the glycan to the slide. Blood group B antigens: (B [type 1]); Lewis antigens: Lewisᵇ (Leᵇ), Lewisᶜ (Leᶜ), and Lewisˣ (Leˣ).

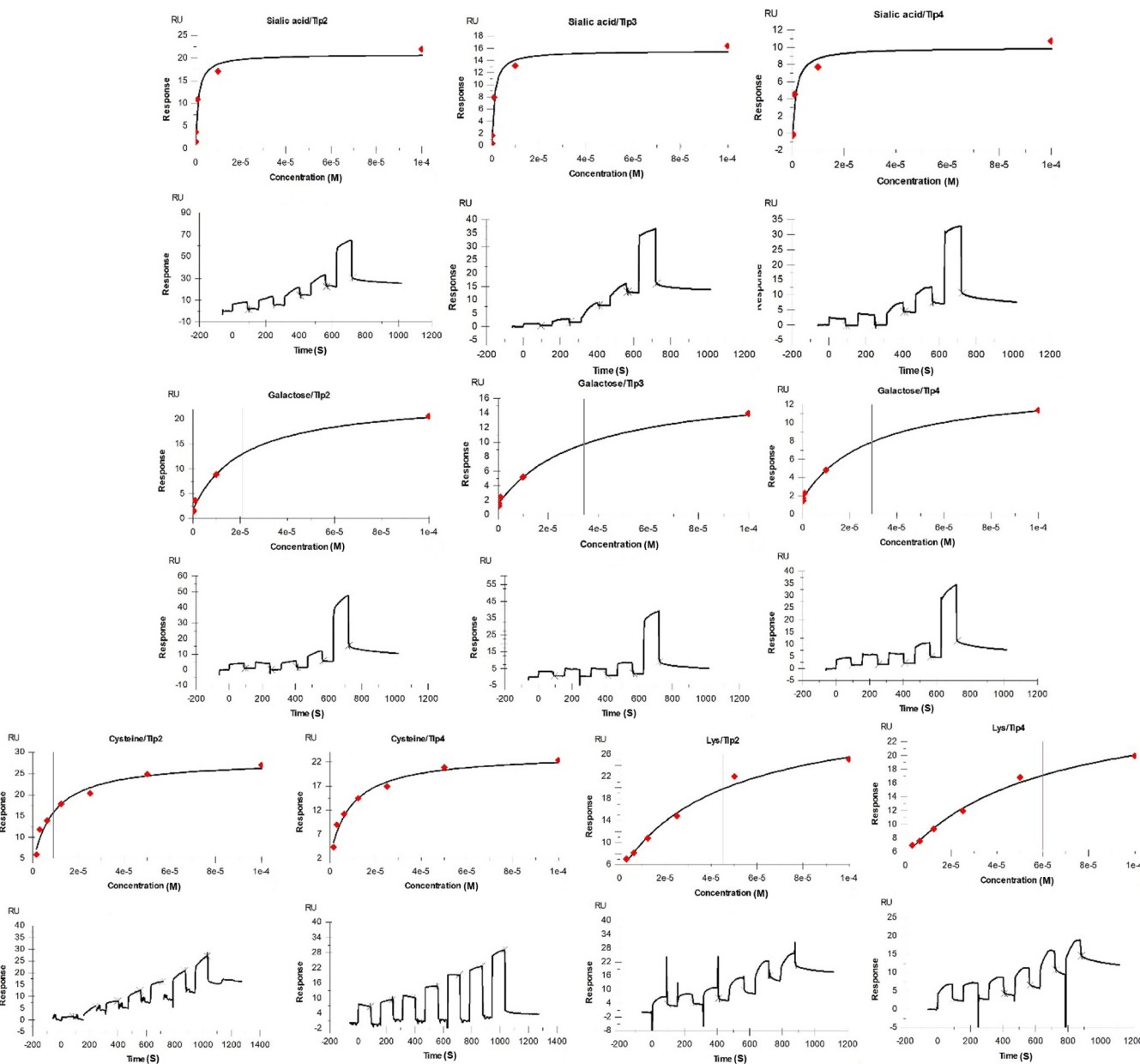

**FIG 4** Representative sensograms from SPR analysis of Tlp3[peri], Tlp2[peri], and Tlp4[peri]. Single-cycle SPR plots are shown as concentration-dependent interactions between proteins and ligands, as illustrated.

additive. When two ligands share one site, the SPR signal is limited to that of a single ligand. Preferential and flexible site binding will produce an intermediate signal.

We interrogated the competitive binding of all ligands to Tlp3[LBD] in the following combinations: (i) glycan versus glycan, (ii) glycan versus amino acid, (iii) glycan versus organic compound, and (iv) amino acid versus amino acid, as exemplified in Fig. 6. The competition interaction of Tlp3[LBD] with glucose versus fucose (Fig. 6A) exhibited no additive response (as measured in response units [RU]), indicating that both ligands compete to occupy a single specific site (the steps are shown in Fig. S6). Tlp3[LBD] also allows preferential binding to a shared site, such as glucose and isoleucine (or arginine) (sugar versus amino acid) or isoleucine and thiamine (amino acid versus amino acid) (Fig. 6B and C), where the two ligands can bind the same site but, when both are present, one outcompetes the other, resulting in preferential binding. Tlp3[LBD] can also bind ligands in a flexible binding; an example of this is the binding of glucose and arginine

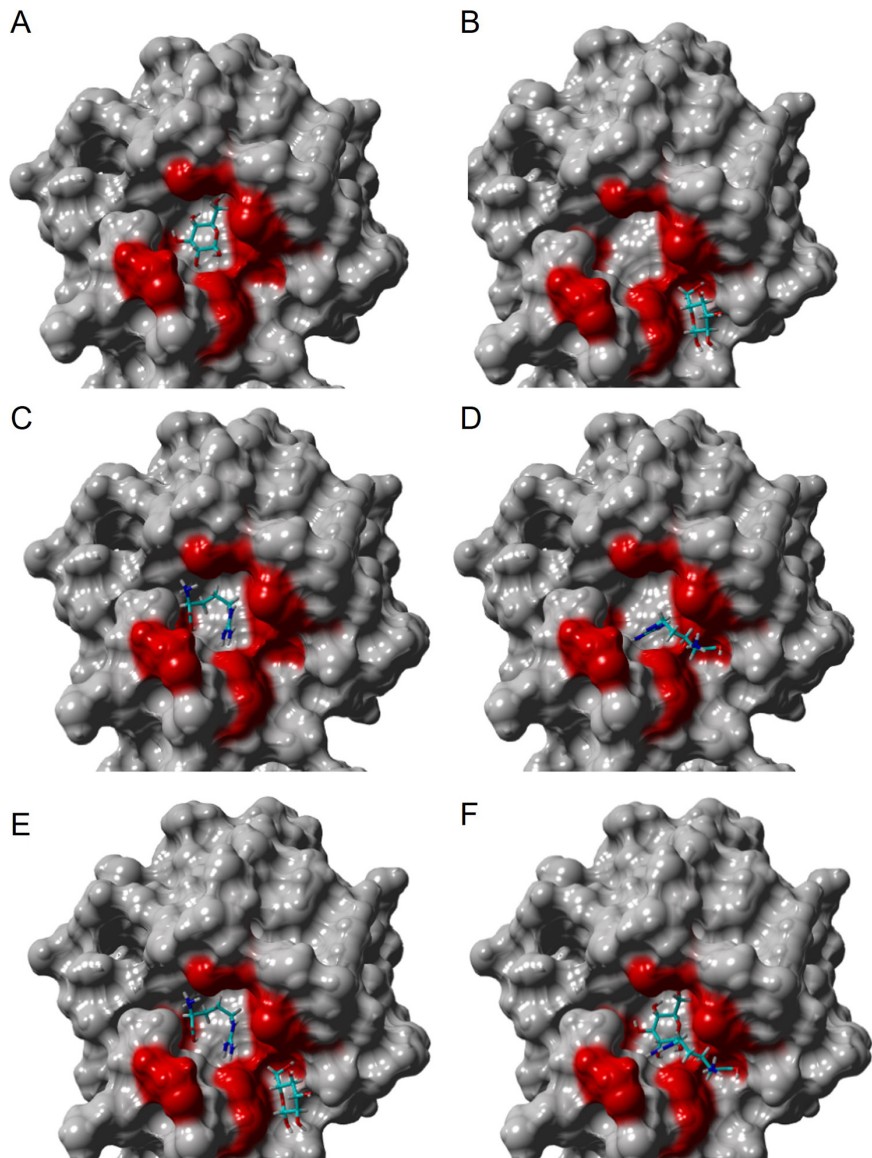

**FIG 5** Molecular docking results of glucose (Glc) and arginine (Arg) bound to cluster A of Tlp3. Glc bound to cluster A's main (A) and extended (B) sites; Arg bound to cluster A's main (C) and extended (D) sites; Arg bound to cluster A's main site and Glc bound to the extended site (E); Glc bound to cluster A's main site and Arg bound to the extended site (F). Structures E and F were submitted to 50-ns Molecular Dynamics (MD) simulations using the AMBER force field (see the supplemental material).

or glucose and thiamine (glycan versus amino acid) (Fig. 6D and E), where glucose could bind to Tlp3[LBD] together with arginine. When glucose is added first, the subsequent addition of arginine cannot replace glucose, and this results in partial summation of the responses, compared to theoretical, indicating the availability of a second site for arginine (or thiamine). However, when arginine is added first, glucose cannot bind, which indicates that both ligands share the same site, indicating that the binding site can accommodate both ligands only when glucose is bound first. This is consistent with the molecular modeling of Tlp3[LBD] with glucose and arginine (Fig. 5; see also Videos S1 and S2 in the supplemental material). These analyses enabled the assignment of all Tlp3 chemoeffectors to the same ligand-binding pocket (cluster A, Fig. 3) with a high-affinity binding site, and a secondary low-affinity binding site that can accommodate arginine and thiamine when the first one was saturated with high-affinity

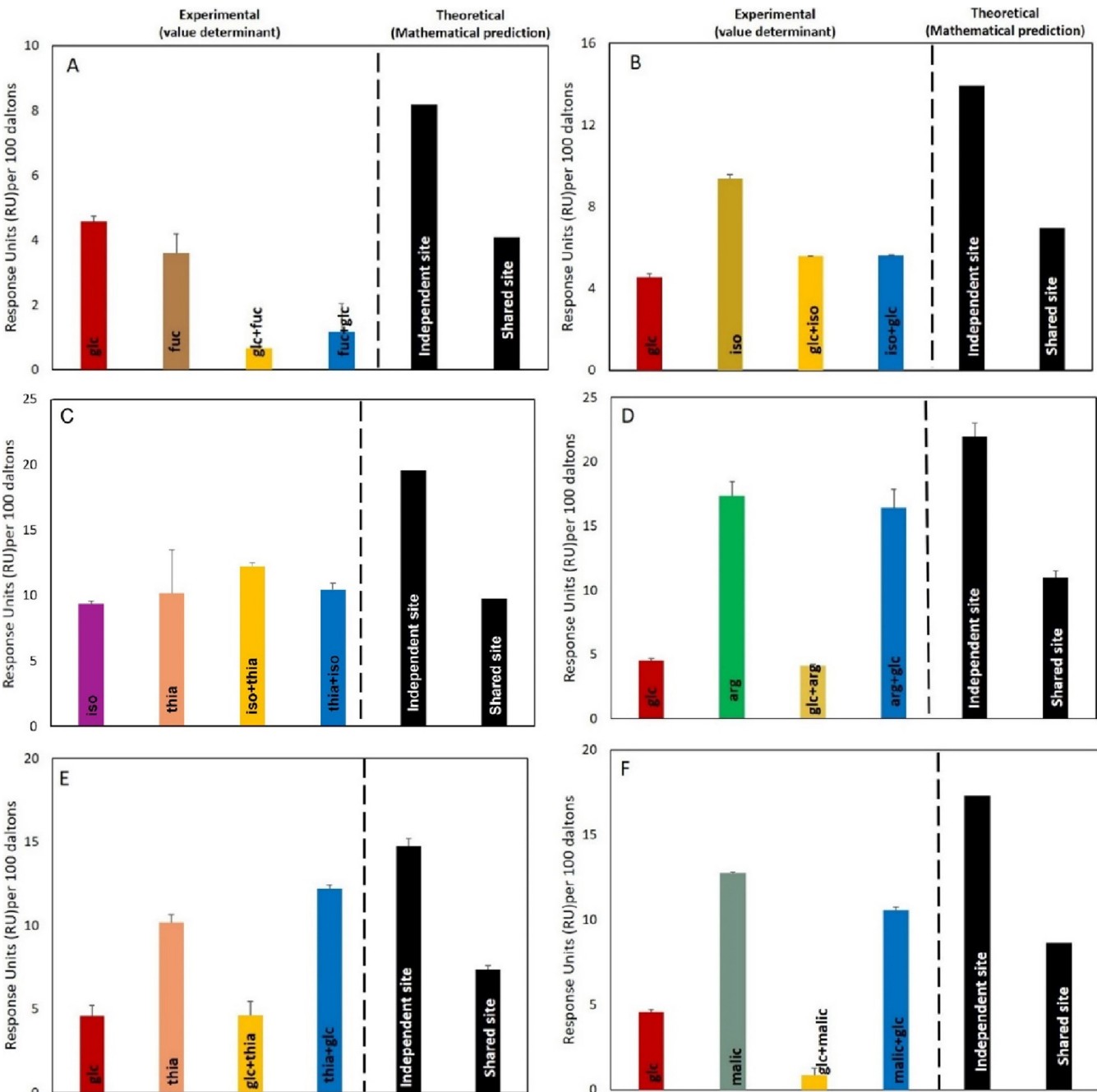

**FIG 6** SPR competition analysis illustrates one binding pocket with two distinct binding sites for Tlp3 ligands. SPR competition analysis of the binding of glucose (glc), fucose (fuc), thiamine (thia), arginine (arg), and isoleucine (iso) to WT Tlp3$^{LBD}$ was performed. The experimental RU value is the actual RU value, as follows. (A) "glc" indicates responses to glucose only, "fuc" indicates responses to fucose only, "glc+fuc" indicates glucose responses following saturation with fucose, and "fuc+glc" indicates fucose responses following saturation with glucose (see Fig. S6 for the representative ABA sensorgram for this analysis). (B) "iso" indicates responses to isoleucine only, "glc" indicates responses to glucose only, "iso+ glc" indicates isoleucine responses following saturation with glucose, and "glc+iso" indicates glucose responses following saturation with isoleucine. (C) "iso" indicates responses to isoleucine only, "thia" indicates responses to thiamine only, "iso+thia" indicates isoleucine responses following saturation with thiamine, and "thia+iso" indicates thiamine responses following saturation with isoleucine. (D) "glc" indicates responses to glucose only, "arg" indicates responses to arginine only, "glc+arg" indicates glucose responses following saturation with arginine, and "arg+glc" indicates arginine responses following saturation with glucose. (E) "glc" indicates responses to glucose only, "thia" indicates responses to thiamine only, "glc+thia" indicates glucose responses following saturation with thiamine, and "thia+glc" indicates thiamine responses following saturation with glucose. (F) "glc" indicates responses to glucose only, "malic" indicates responses to malic acid only, "glc+malic" indicates glucose responses following saturation with malic acid, and "malic +glc" indicates malic acid responses following saturation with glucose. The theoretical value is RU values based on mathematical theory. The binding status of the ligands to protein are classified as follows: independent site (additive/accumulative effect), ligands binding to different binding sites; shared site, ligands binding/sharing same binding site; or preferential shared site, ligands binding/sharing same binding site, but the protein binds to one ligand better than the other when in equilibrium. All response data were normalized to a 100-Da molecular weight for each analyte, allowing direct comparison of responses.

ligands (see Fig. S7). Overall, the docking and competitive SPR data support the hypothesis that the ligands bind to Tlp3$^{LBD}$ via one flexible pocket containing high- and low-affinity binding sites.

**Inactivation of sensory LBDs of *tlp2*, *tlp3*, and *tlp4* affects chemotactic motility of *C. jejuni*.** The biological relevance of the binding specificities of the Tlp2,3,4 cluster was assessed by examining chemotactic responses of *C. jejuni* 11168-O *tlp2*, *tlp3*, and *tlp4* wild-type (WT) and mutant strains using a nutrient-depleted chemotaxis assay. All ligands, with the exception of lysine and cysteine, interacted with Tlp2 and Tlp4 as attractants (Table 1), showing an ~1- to ~2-log decrease ($P < 0.05$) in migration toward the ligand compared to the WT strain. Lysine was found to be a repellent for Tlp2, exhibiting a 1.4-log increase ($P < 0.05$) in migration, while cysteine did not generate any significant chemotactic response ($P < 0.05$) (Fig. 7). Interestingly, mannose and sialic acid appeared to act as attractants for Tlp2 and Tlp4 but as repellents for Tlp3.

**Collective inactivation of sensory LBDs of *tlp2*, *tlp3*, and *tlp4* further alters the chemotactic motility of *C. jejuni*.** To further elucidate Tlp2,3,4 mediated chemotaxis of *C. jejuni*, three double ($\Delta tlp2,4$, $\Delta tlp2,3$, and $\Delta tlp3,4$) and a triple isogenic inactivation mutant strain ($\Delta tlp2,3,4$) were interrogated. The chemotactic response to a ligand that is either an attractant or a repellent for all three Tlps yielded a cumulative response, exemplified by the $\Delta tlp2,3,4$ triple mutant showing the greatest reduction in migration toward purine (Fig. 7). However, when a ligand acted as an attractant for one of the Tlps but as a repellent for others, a more complex chemoresponse was observed. Lysine, for example, was a repellent for Tlp2 and Tlp3 (11) but an attractant for Tlp4. Cell migration toward lysine was increased by 1.8-log ($P < 0.05$) for the $\Delta tlp2,3$ mutant and decreased by 1.1-log ($P < 0.05$) for the $\Delta tlp2,4$ mutant, but there was no change in migration observed for the $\Delta tlp3,4$ mutant. The $\Delta tlp2,3,4$ triple mutant showed an increase of 1.5-log ($P < 0.05$) in cell migration (Fig. 7). This indicates that chemoresponses are not always cumulative and are likely to be the result of a complex balance in response to attractant and repellent stimuli. Migration toward mucin, used as a control, was not affected by mutations. Results of the nutrient depletion assay were confirmed by the modified tHAP chemotaxis assay (34) (see Fig. S8 and Table S5).

**The conserved amino acids in the binding pocket of Tlp3$^{LBD}$ are involved in ligand interaction.** Our modeling analysis showed that K149, W151, D169, T170, and D196 are part of the main site of the binding pocket of Tlp3$^{LBD}$ (Fig. 2; see also Fig. S3). These residues were also shown by crystallography to interact with isoleucine for Tlp3 and play direct role in this interaction (15). K149, W151, D169, D196, and T170 amino acid residues were replaced with alanine (A) by site-directed mutagenesis to create five single-residue mutants: Tlp3$^{LBD \ K149A}$, Tlp3$^{LBD \ W151A}$, Tlp3$^{LBD \ D169A}$, Tlp3$^{LBD \ T170A}$, and Tlp3$^{LBD \ D196A}$. SPR analysis was then used to determine the change in the binding affinities of these site-directed mutants for all Tlp3 ligands (Table 3). Single-residue substitutions for all five mutated Tlp3$^{LBD}$s led to significant reduction ($P < 0.05$) of the binding affinity of each protein for arginine (1- to 3-fold), isoleucine (3- to 5-fold), lysine (3- to 7-fold), fucose (1 to 2-fold), glucosamine (3- to 6-fold), glucose (1- to 2-fold), and sialic acid (1- to 4-fold) (Table 3). This further confirms that these ligands interact with a single binding pocket in Tlp3$^{LBD}$.

Substitutions K149A, W151A, and D169A exhibited 2-fold ($P < 0.05$) reductions in the affinity of Tlp3$^{LBD}$ for a single monosaccharide, mannose. The substitution W151A led to reductions in binding by only 1-fold ($P < 0.05$) to aspartate, 3-fold ($P < 0.05$) to malic acid, and 1-fold ($P < 0.05$) to thiamine compared to WT Tlp3$^{LBD}$ (Table 3).

## DISCUSSION

Here, we report that a unique paralogous group of dCache_1 chemoreceptor proteins—Tlp2, Tlp3, and Tlp4 of *C. jejuni*—possess the ability to respond to a wide range of effectors, including organic/amino acids, as well as simple and complex glycans. This distinctive ability to respond to both glycans and organic and amino acids by dCache_1 proteins has not been previously reported for other sensory proteins of

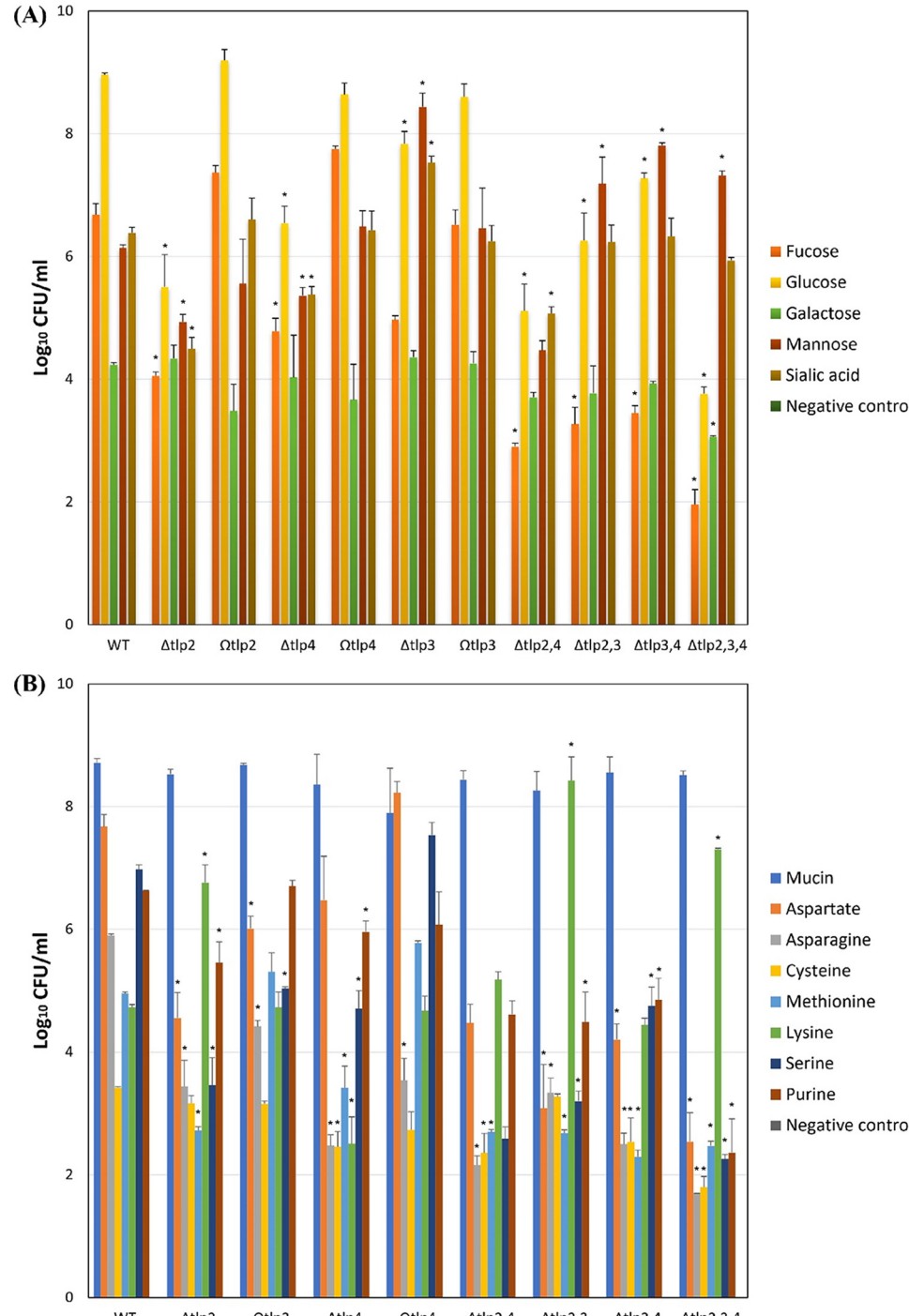

**FIG 7** Nutrient depletion chemotaxis assay. The agarose plugs contained the indicated ligands. (A) fucose, glucose, galactose, mannose, and sialic acid. (B) Serine, methionine, asparagine, lysine, cysteine, purine, and aspartate. The *C. jejuni* 81116 Δ*flaA* Δ*flaB* isogenic mutant was used as a nonmotile, nonchemotactic control; agar plugs containing no added ligand were used as a negative control, and mucin was used as a positive control. Standard errors are shown as bars above the means of three replicates. Viable counts of *C. jejuni* from the assays are shown on a log scale. The asterisk (*) indicates a statistically significant difference compared to the WT strain ($P < 0.05$).

dCache_1 family. Sequence analysis of Tlp2, Tlp3, and Tlp4 sensing domains revealed that all three possess an AA_motif (27).

Phylogenetic analysis of dCache_1-containing chemoreceptors from representatives of the *Campylobacterota* phylum showed that nearly all genomes encode at least one paralogous amino acid-sensing chemoreceptor. Many species possess as many as

**TABLE 3** Binding affinity ($\mu$M) of Tlp3$^{LBD}$ and Tlp3$^{LBD}$ mutants to ligands[a]

| Ligand | Mean ± SD | | | | | |
|---|---|---|---|---|---|---|
| | Tlp3$^{LBD\ WT}$ | Tlp3$^{LBD\ K149A}$ | Tlp3$^{LBD\ W151A}$ | Tlp3$^{LBD\ D169A}$ | Tlp3$^{LBD\ D170A}$ | Tlp3$^{LBD\ D196A}$ |
| Lysine (Lys) | 4.3 ± 1.6 | 32.3 ± 5.3* | 27.3 ± 5.9* | 29.2 ± 4.9* | 14.5 ± 4.8* | 16.1 ± 9.1* |
| Purine (Pur) | 37.2 ± 17.4 | 41.9 ± 24 | 36.1 ± 15.7 | 15.7 ± 6.6 | 20.6 ± 4 | 35.8 ± 1.5 |
| Aspartate (Asp) | 152.3 ± 5.7 | 210.5 ± 38.9 | 245.1 ± 36.6* | 216 ± 3.9* | 234.6 ± 4* | 121 ± 8.1 |
| Malic acid (Mal) | 17.9 ± 4.2 | 34 ± 14.1 | 62 ± 9.7* | 22 ± 9.7 | 25 ± 8.4 | 63 ± 6.3* |
| Arginine (Arg) | 35.3 ± 2.7 | 64 ± 7.1* | 95.4 ± 27.7* | 82.6 ± 17.7* | 56.2 ± 27.6 | 59 ± 11.6* |
| Isoleucine (Iso) | 16.4 ± 1.7 | 82 ± 17.8* | 93 ± 4.7* | 76 ± 18.5* | 61 ± 13.6* | 54 ± 16.1* |
| Glucosamine (GlcN) | 7.1 ± 2.9 | 24.3 ± 2.1* | 34.3 ± 12.7* | 38.4 ± 6.45* | 39.7 ± 8.6* | 45.5 ± 7.3* |
| Thiamine (Thia) | 41.4 ± 5.9 | 53.8 ± 6.6 | 73.3 ± 4.4* | 62 ± 8.5* | 30.6 ± 7.2 | 36.8 ± 10 |
| Glucose (Glc) | 58.2 ± 9.7 | 113.2 ± 6.6* | 126.8 ± 4.4* | 94.2 ± 8.5* | 80.6 ± 4* | 67.8 ± 8.7* |
| Fucose (Fuc) | 25.2 ± 6.5 | 63.7 ± 16.8* | 66.9 ± 14.6* | 50 ± 17.6* | 46.6 ± 16.1* | 41.9 ± 13.2* |
| Galactose (Gal) | 45.1 ± 1.3 | 53.2 ± 5.1 | 56.4 ± 9.7 | 64.9 ± 14.1 | 42.6 ± 11 | 64.4 ± 19.3 |
| Mannose (Man) | 21.2 ± 2.4 | 39.8 8.5* | 37.3 ± 4.9* | 43.6 ± 2.1* | 35.9 ± 9 | 26.8 ± 6.1 |
| Sialic acid (Sia) | 9.0 ± 0.2 | 24 ± 6.6* | 17.3 ± 4.4* | 25.2 ± 8.5* | 16.6 ± 2* | 36.8 ± 3* |

[a]Data represent the means of three independent experiments ($n = 3$). *, Significant differences in binding affinity ($P < 0.05$) compared to that of the wild type (Tlp3$^{LBD\ WT}$).

six paralogs, and the pattern of distribution of these across phylogenetic tree revealed that they were the result of multiple independent gene duplication events. A substantial number of *C. jejuni* strains lack Tlp2,3,4-type chemoreceptors, and we discovered that in some genomes, where these receptors are absent, the genes encoding them have become pseudogenes.

Our analyses suggest that *tlp2* and *tlp4* are the result of the gene duplication that happened prior to the divergence of the *Campylobacteraceae* and *Helicobacteraceae* families. Tlp3 appeared more recently through the *tlp2* gene duplication before *C. jejuni* and *C. coli* divergence. Expectedly, upon duplications, the sensory regions of all three receptors underwent considerable changes, leading to the low sequence identity between them (~30%). Moreover, significant changes also occurred in the AA_motifs of Tlp2 and Tlp3. Whereas in Tlp4 the consensus AA_motif was preserved, tryptophan in Tlp2 and the first tyrosine and arginine in Tlp3, which are all essential for recognizing amino acid ligand carboxyl groups (27), are substituted by tyrosine, leucine, and lysine, respectively (Fig. 2).

Gene duplication can result in two distinct outcomes if both gene copies are retained in the genome: neofunctionalization and subfunctionalization. For example, three paralogous amino acid-sensing chemoreceptors from *P. aeruginosa* (PctA, PctB, and PctC) evolved to recognize different amino acid repertoires (26). Strikingly, despite the differences in the AA_motif, the *C. jejuni* Tlp2 and Tlp4 ligand-binding specificities appear to be almost identical, and their ligand repertoire is exceptionally broad. It would be valuable to determine the structural basis of Tlp2 and Tlp4 specificity.

*C. jejuni* is predominantly an asaccharolytic organism, with the exception of a few strains that can sense fucose (35, 36). *C. jejuni* 11168-O can catabolize fucose, but it does not encode the genes for glucose catabolism (37, 38); however, our data show that, in addition to fucose, Tlp2,3,4 can sense galactose and glucose as attractants and initiate chemotaxis toward these effector molecules. Therefore, it is possible that some of the Tlp2,3,4 sensory abilities may contribute to the relocation of the bacteria to more favorable conditions or to target specific host tissues.

The ability of Tlp2,3,4 to bind multiple ligands with a broad and overlapping sensory repertoire could explain why many *C. jejuni* strains lack either one, two, or even all three of the *tlp2,3,4* genes. The ligand-sensing redundancy in the *C. jejuni* chemosensory repertoire is evident not only in the overlapping ligand specificities of Tlp2,3,4 but also in its presence among other receptors (8, 10, 11). This has likely given *C. jejuni* a possible evolutionary advantage against gene loss and allowed adaptation to complex and varied environments, such as are found in water reservoirs, ponds, and human and avian intestinal tracts.

It is interesting to note that some of the effectors act as attractants for one receptor and repellents for others, as exemplified by the sensory specificity of Tlp2,3,4 to lysine. Tlp2 and Tlp3 sense lysine as a repellent, and Tlp4 senses it as an attractant. Deletion

of Tlp2 and Tlp3 LBD resulted in an increased migration of the isogenic mutant toward lysine, which is consistent with Tlp4, as the only remaining receptor of the group, responding to lysine as an attractant. However, when all three receptor sensory domains were removed, the strain still exhibited increased migration toward lysine (Fig. 7). This is possibly due to a yet-to-be-discovered sensory redundancy for lysine or a receptor-periplasmic protein pair that could navigate the bacteria toward lysine, similar to the *Bacillus subtilis* multi-ligand-binding dCache_1 receptor, McpC. McpC appears to be involved in indirect binding to lysine via ancillary proteins to facilitate bacterial movement (39). It has been previously suggested that a combination of receptors could sense the same chemoeffector as an attractant and a repellent, thereby aiding the microorganisms in evading harmful concentrations of some chemoeffectors (10), and it has been shown that bacteria are able to balance between attractant and repellent signals in order to find an optimum environment with sensitivity to repellents being greater than that to attractants (40). The complex interaction of chemoreceptors, the formation of different trimers-of-dimers, and the direct interaction of other companion proteins in the chemotaxis signaling system under different growth conditions has been reported in *Escherichia coli* (7, 41–43). It is therefore highly plausible that the chemoreceptors in *C. jejuni* work together as well to interact with other receptor/regulatory proteins to ensure that the bacteria find the most favorable niche.

Despite high affinity for cysteine ($K_D$ 10.8 $\mu$M; Table 1), removal of Tlp2 sensing did not affect the chemotactic movement of this isogenic strain, indicating a different relationship of this molecule with Tlp2, possibly as an antagonist. The evidence for some ligands as antagonistic inhibitors of receptors, or pathway inhibitors, is beginning to emerge. Chemosensor antagonists have been reported for *P. aeruginosa* (44) and *Helicobacter pylori* (33).

The dCache_1 domain chemoreceptors can bind ligands via both membrane-distal and -proximal binding pockets such as in TlpA and TlpC receptors in *H. pylori* (33, 45). Interaction of multiple ligands via a single pocket in the membrane-distal domains has also been reported for several dCache domain multi-ligand-binding receptors, such as McpX/Mlp24 from *Vibrio cholerae* (46), McpB from *B. subtilis* (39), McpU in *Sinorhizobium meliloti* (47–49), and PctA, PctB, and PctC of *P. aeruginosa* (26). In Tlp3$^{LBD}$, the distal binding pocket was shown to have strong binding interactions with its ligands, and substitution of five conserved amino acids, located in the distal pocket, with alanine adversely affected the binding of the ligands. Considering the amino acid sequence similarity and conservation of the ligand-binding motif in Tlp2,3,4 sensory domains, it is possible to postulate that Tlp2 and Tlp4 interacts with ligands in a similar manner to Tlp3 and that these cluster-specific conserved amino acids are involved in the direct binding of ligands. The diversity of the adjacent amino acids around these conserved clusters is likely to contribute to the recognition of different ligands and to the variability observed in the sensory domains of the MCP family (50).

Furthermore, molecular modeling, STD NMR and competition SPR presented here demonstrate that Tlp3 has a single but extended ligand-binding pocket in the membrane-distal domain. This pocket has one main site and an adjacent secondary site. These sites appear to be able to accommodate more than one chemoeffector simultaneously. It is possible that different amino acid residues in the distal ligand binding pocket(s) of Tlp3 may serve as specific anchor residues for different ligands in order to accommodate such diverse molecules either singly or together. This is supported by the evidence of reduced binding affinity of Tlp3$^{LBD}$ to some ligands, such as arginine, but not others, such as glucose, when the conserved amino acid binding residues K149, W151, and D169 were mutated to alanine. It is yet to be determined whether these sites (main and extended) can allosterically affect one another or how signaling could be affected by the simultaneous binding of the attractants and repellents.

It is also noteworthy that the sensory repertoire all three receptors—Tlp2,3,4—is consistent with the previous report by Rhaman et al. (11) but contradicts the report by Khan et al. (51). This is possibly due to several factors, including the pH of the buffers,

that can adversely alter experimental receptor-ligand interactions (52). Most importantly, the use of Tris-based buffers in the ligand interaction assays by Khan et al. (51) can structurally mimic amino acids, since Tris is an organic buffer [2-amino-2-(hydroxymethyl)-1,3-propanediol] and has three -OH groups and an -NH$_2$ group. Tris buffers are normally used for stabilization of purified proteins. However, Tris molecules have been extensively reported to be the source of interference in protein-ligand interaction assays (53–55).

*C. jejuni* can inhabit complex niches and thereby has the ability to sense and bind to various glycan structures found in mucus, which includes terminal fucose, glucose, galactose, mannose, and sialic acid (56, 57). Tlp2,3,4 showed interaction with a wide range of glycosylated structures, including blood group antigens such as H-type, Lewis's antigens such as Le$^a$, Le$^x$, and Le$^y$, and fucosyl GM1 ganglioside (Table 2; see also Table S3). Interestingly, Tlp2,3,4 exhibited interactions with P antigens (see Table S3), including P1 and P$^k$ antigens, which are clinically important in disease association and progression. P antigens serve as receptors for P-fimbriated pathogenic *Escherichia coli* which affects epithelial cells of the human urinary tract and cause chronic infections (58, 59). Fucose residues are commonly found in the terminal positions of glycosylated mucin proteins, in digestive tracts, and on cell surfaces and are important to the intestinal microflora (36, 60, 61). Mucin proteins in particular, such as cell surface MUC1 and secreted Muc2, are ubiquitously present in the human intestine, and we have previously reported that the *C. jejuni* chemoreceptors Tlp10 and Tlp11 can modulate greater adhesion of campylobacteria to human cell culture expressing MUC1 proteins (10, 13). The relationship between infectious microorganisms and fucose is well established, and *C. jejuni* has been reported to upregulate genes, e.g., flagellar genes that play essential roles in colonization and chemotaxis (62, 63). Glycans typically play crucial roles in the colonization, invasion, and biofilm formation of *C. jejuni* (64–66).

It is becoming evident that chemoreceptors work in concert and that the final signal output is likely to be influenced not only by the environmental cues but also by metabolic output, cytoplasmic redox sensors, and axillary proteins that facilitate recognition of a wide range of effectors. A more detailed interrogation of these complex interactions between chemoreceptors, chemoeffectors, and other sensory components is needed to further understand the role of chemosensing in the pathogenicity of *C. jejuni* and related organisms.

## MATERIALS AND METHODS

**Microbial strains, growth conditions, and plasmids.** All bacterial strains, growth conditions, and plasmids used in this study are listed in Table S6. The *C. jejuni* 11168-O strain was kindly provided by Diane Newell from Central Veterinary Laboratories, London, UK. All *C. jejuni* strains were grown microaerobically at 42°C (5% O$_2$, 10% CO$_2$, 85% N$_2$) for 18 to 24 h on Mueller-Hinton agar/broth (MHA/MHB; Oxoid) which was supplemented with vancomycin (10 $\mu$g mL$^{-1}$) and trimethoprim (2.5 $\mu$g mL$^{-1}$). For mutational studies, *E. coli* strains with plasmids were grown at 37°C in Luria-Bertani (LB) medium (Oxoid) with ampicillin (100 $\mu$g mL$^{-1}$), kanamycin (50 $\mu$g mL$^{-1}$), and chloramphenicol (30 $\mu$g mL$^{-1}$) as required.

**Expression and purification of the sensory domains of Tlp2, Tlp3, and Tlp4.** His-tagged proteins (Tlp2, -3, and -4) were prepared as previously described (11) DNA segments encoding the periplasmic sensory domains (33 to 286 amino acids) of Tlp2 and (38 to 297 amino acids) of Tlp4 were amplified (see Table S7), ligated into pGEM-T Easy (Promega) and subcloned into pET-19b (Novagen) (see Table S6). Recombinant proteins were expressed in *E. coli* BL21(DE3) using 1 mM IPTG (isopropyl-$\beta$-D-thiogalactopyranoside) and purified using His-select-nickel chromatography as described previously (8, 11). The purity of the proteins was confirmed by SDS-PAGE and Western blot analysis with anti-His antibodies (Bio-Rad). Protein concentrations were determined using a NanoDrop spectrophotometer (Thermo Fisher).

**Identification of ligand interactions of Tlp2, Tlp3, and Tlp4 by glycan/small molecule arrays.** Glycan (>400 monosaccharides or complex glycans) and small molecule arrays containing amino acids and salts were performed as described previously by Day et al. (67). Briefly, 1 $\mu$g of purified recombinant protein was complexed with anti-His antibody (Cell Signaling Technology) and rabbit anti-mouse/goat anti-rabbit Alexa 555-IgG (Thermo Scientific) at a molar ratio of 1:0.5:0.25 and then incubated on the array for 30 min in phosphate-buffered saline (PBS). The array slides were scanned at 488/520 nm and analyzed using InnoScan 1100 with MAPIX software. All arrays were performed in triplicate with a total of 12 data points for each glycan tested; the complete data set and MIRAGE (i.e., minimum information regarding a glycomics experiment) results are shown in Tables S3 and S8.

**Surface plasmon resonance analysis.** Surface plasmon resonance (SPR) experiments were performed as described previously (10, 11) with the following modifications. Briefly, purified proteins were

diluted to 0.15 $\mu$M in sodium acetate (pH 4.0) and immobilized onto an active flow cell of a CM5 sensor chip with a flow rate of 5 $\mu$L/min for 7 min of contact time using a standard amine coupling protocol. Immobilized protein levels for all three proteins ranged from 6028.3 to 9779.3 response units (RU) providing a minimum expected response maximum (RMax) between 20 and 50 RU for the small molecules tested. Flow cell 1 was used as a reference that contained no ligand and was only blocked with ethanolamine. All ligands were prepared in PBS (pH 7.2) and serially diluted from 0.2 to 0.0125 mM. A 10-min dissociation period was allocated after the introduction of each analyte. SPR experiments were performed in single-cycle kinetics mode using the BIAcore S200 biosensor system (Cytiva) at 25°C in PBS (pH 7.2) at a flow rate of 30 $\mu$L/min. The equilibrium dissociation constant ($K_D$) for each analyte was determined using BIAcore S200 Evaluation software with a report point 4 s after the injection stop to account for the bulk transfer of the small molecule against a large amount of immobilized protein (as shown in Fig. S5).

**SPR competition assays.** SPR competition assays were performed via the A-B-A inject function on a BIAcore S200 instrument as described by Elgamoudi et al. (10). Competition A-B-A analyses determine the specificity of the potential ligand-binding site preferences of WT Tlp3$^{LBD}$. The bound protein is saturated with the first analyte (A), followed by the introduction of a second analyte (B); the assay is designed to show whether a cumulative response is observed after the second analyte (B) is added. A cumulative response indicates two independent binding sites, whereas the lack of a cumulative response indicates a single site where ligands compete for binding, i.e., when analyte A occupies a ligand binding site, analyte B cannot bind. Since all analytes are used at saturation, this assay does not provide 1:1 competition to indicate the identity of the preferred analyte. The A-B-A approach was used with combinations of each of the compounds (at a concentration 10 × $K_D$) and PBS controls, with 60-s injections of analyte A to ensure that saturation or near saturation was reached prior to competition with analyte B. The results were analyzed using BIAcore S200 evaluation software in the sensorgram mode, and data were zeroed to baseline prior to the initial A injection. All response data were normalized to a 100-Da molecular weight for each analyte, allowing direct comparison of the responses. Both actual RU values and theoretical values are recorded for comparison.

**Circular dichroism spectroscopy.** Integrity of the purified proteins was confirmed by circular dichroism (CD). CD was performed as described previously (68) using a JASCO J-810CD spectropolarimeter under constant nitrogen flow connected to a Peltier temperature controller. Far-UV spectra were generated at 190 to 260 nm at 4°C in 10 mM PBS. Spectra were obtained using a 0.1-cm path length and a protein concentration of 0.1 mg/mL. Buffer scans were subtracted from the protein scans, and the data were smoothed using JASCO software. In all experiments, the high-tension voltage was kept below 600 V. The far-UV CD spectra were used to verify the integrity of the purified periplasmic domains of Tlp2, Tlp3, and Tlp4 (see Fig. S9).

**STD NMR spectroscopy.** STD NMR experiments were performed as previously described in (10). Briefly, recombinant Tlp3 (1 mg/mL) was buffer exchanged into 50 mM NaCl and 50 mM KH$_2$PO$_4$ in D$_2$O (99.99% D Cambridge Isotopes) containing a 100-times molecular access of arginine (Arg), resulting in a total volume of 200 $\mu$L. A control sample was prepared in an identical manner without the addition of Tlp3. All NMR experiments were performed using 3-mm NMR tubes on a Bruker Avance 600 MHz spectrometer equipped with a 5-mm TXI probe with triple-axis gradients at 283K without sample spinning. $^1$H NMR spectra were acquired with 32 scans and a 2-s relaxation delay over a spectral width of 6,000 Hz. Solvent suppression of the residual HDO peak was achieved by using a continuous low-power presaturation pulse during the relaxation delay. For all STD NMR experiments, the protein was saturated at −0.5 ppm in the aliphatic region of the spectrum and off-resonance at 33 ppm with a cascade of 40 selective Gaussian-shaped pulses of 50-ms duration (50 dB), which correlates to a strength of 190 Hz. A 100-$\mu$s delay between each soft pulse was applied, resulting in a total saturation time of ∼2 s 2 K scans were acquired per sample. Data were obtained with interspersed acquisition of pseudo-two-dimensional on-resonance and off-resonance spectra in order to minimize the effects of temperature and magnet instability. On- and off-resonance spectra were processed separately, and the final STD NMR spectrum was obtained by subtracting the individual on- and off-resonance spectra, resulting in low-subtraction artifacts.

**Mutation and complementation of *tlp2* and *tlp4* in *C. jejuni* 11168-O.** The *tlp2* and *tlp4* coding region were amplified from the *C. jejuni* 11168-O strain using forward (5′-CATATGAAAAGCGTAAAATTG-3′) and reverse (5′-CTCGAGTTAAAACCTCTTCTTCTTAACATC-3′) primers for *tlp2* and forward (5′-GGTACCC AATCAATAAATTCAGG-3′) and reverse (5′-CTCGAGTTAAAAACCTCTTCTTCTTAACATC-3′) primers for *tlp4* (see Table S7). The PCR fragments were cloned into pGEM-T Easy to produce the intermediates pGU0504 and pGU0706, respectively (see Table S6). Then, the genes were inactivated by inverse PCR mutagenesis (11), where primers were designed to incorporate a BglII restriction site (see Table S7) to delete 925 bp of *tlp2* (from 582 to 1507 bp) and to delete 747 bp of *tlp4* (from 45 to 792 bp). Nonpolar erythromycin resistance cassette (for *tlp2*) and chloramphenicol resistance cassette (for *tlp4*) were inserted in the same sense orientation as the genes to generate pGU0510 and pGU0720, respectively (see Table S6). The isogenic *tlp2* and *tlp4* mutant strains, 11168-OΔ*tlp2*::Er$^r$ and 11168-OΔ*tlp4*::Cm$^r$, respectively, were constructed by electrotransformation of recombinant plasmids into the *C. jejuni* 11168-O strain. The integration of the mutant allele was verified by PCR and DNA sequencing (GUDSF, Australia). The mutants were complemented by the insertion of *tlp2* and *tlp4* full-length genes. Briefly, *tlp2* and *tlp4* genes were amplified via PCR to include BsmBI restriction sites, cloned into pGEM-T Easy (Promega) and subcloned into pK46 suicidal vector as described by Rahman et al. (11). The pK46 harboring *tlp2* (pK46*tlp2*) or *tlp4* (pK46*tlp4*) were cloned into 11168-OΔ*tlp2*::Er$^r$ and 11168-OΔ*tlp4*::Cm$^r$ strains, respectively, using natural transformation. Potential recombinants were isolated by growth on 20 $\mu$g/mL erythromycin and 50 $\mu$g/mL kanamycin (for *tlp2*) and

10 $\mu$g/mL chloramphenicol and 50 $\mu$g/mL kanamycin (for *tlp4*). The complemented strains of 11168-O$\Delta$*tlp2*::Er$^r$ $\Omega$*tlp2*::Km and 11168-O$\Delta$*tlp4*::Cm$^r$ $\Omega$*tlp4*::Km were verified by PCR and sequence analysis (GUDSF).

**Generation of double and triple isogenic mutants of *tlp2*, *tlp3*, and *tlp4* in *C. jejuni* 11168-O.** Plasmids harboring inactivated genes—i.e., kanamycin inserted *tlp3* (pGU0817; Rahman et al. [11]), erythromycin inserted *tlp2* (pGU0510), and chloramphenicol inserted into *tlp4* (pGU0720)—were transformed naturally into the *C. jejuni* 11168-O strain. This generated $\Delta$*tlp2,3*, $\Delta$*tlp2,4*, $\Delta$*tlp3,4*, and $\Delta$*tlp2,3,4* strains. Added mutations were verified by PCR amplification and DNA sequencing (GUDSF). All mutants were grown on MHA supplemented with appropriate combination of the antibiotics erythromycin (50 $\mu$g mL$^{-1}$), kanamycin (50 $\mu$g mL$^{-1}$). and chloramphenicol (30 $\mu$g mL$^{-1}$).

**Site-directed mutagenesis.** Single-point alanine substitutions were introduced at positions K149, W151, D169, T170, and D196 via oligonucleotide-directed mutagenesis using a QuikChange kit (Stratagene) as described by the manufacturer.

**Chemotaxis assays.** Chemotaxis assays—the nutrient-depleted chemotaxis assay and the modified hard agar plug assay (tHAP)—were performed as previously described (11, 34). The chemotactic responses of all isogenic strains were compared to that of the *C. jejuni* 11168-O wild-type strain to distinguish between chemoattractants and chemorepellents. For the nutrient-depleted chemotaxis and modified hard agar plug (tHAP) assays, all differences were calculated as changes on a log$_{10}$ scale. Controls were provided by using nonmotile (*C. jejuni* 81116 *flaAB*) and nonchemotactic (*C. jejuni* NCTC 11168 *cheY*) mutants (69) and mucin as negative and positive controls (10).

**Bioinformatic analysis.** Protein sequences were collected from MiST3 database (22). Taxonomy tree and information for phyletic distribution were retrieved from AnnoTree (24) and the Genome Taxonomy Database (21). Multiple sequence alignments were constructed using E-INS-i and L-INS-i algorithms in MAFFT (70). Sequence alignments were edited in Jalview (71). Maximum-likelihood phylogenetic trees were constructed using the MEGA X package (72). Phylogenetic trees were analyzed using the iTOL online tool (v5.5.1) (73).

**Molecular docking analysis.** To identify potential binding sites of Tlp3, blind molecular docking experiments were carried out using AutoDock Vina (28). The Tlp3 X-ray structure (PDB 4XMQ) was used for docking (15). The protein consists of a dimer with two identical structures A and B. For docking experiments, only the monomeric structure was used allowing a global unbiased with a box size of 72.5 $\times$ 72.5 $\times$ 72.5 Å covering the entire protein surface.

**Statistical analysis.** Statistical significance of data generated in this study was determined using two-tailed Student *t* test in Prism (GraphPad Software). A *P* value of $\leq$0.05 was considered statistically significant.

**Data availability.** All data needed to evaluate the conclusions presented here are available within the present study and its associated supplemental material. All materials will be made available on request; an MTA would be required for transfer of recombinant plasmids. Glycan and small molecule arrays are subject to manufacturing costs.

## SUPPLEMENTAL MATERIAL

Supplemental material is available online only.

**SUPPLEMENTAL FILE 1**, PDF file, 2.6 MB.

**SUPPLEMENTAL FILE 2**, MOV file, 9.6 MB.

**SUPPLEMENTAL FILE 3**, MOV file, 8 MB.

## ACKNOWLEDGMENTS

This study was supported in part by the U.S. National Institutes of Health (grant R35GM131760 to I.B.Z.) and in part by the Institute for Glycomics, Griffith University, Gold Coast, Queensland, Australia.

V.K. conceived and designed the study. H.R.T., B.A.E., T.H., C.D., R.M.K., H.R., L.E.H.-T., and T.N. performed the experiments. E.P.A. and I.B.Z. performed bioinformatics analysis. V.K., T., B.A.E., C.J.D., E.P.A., T.H., and I.B.Z. analyzed the data and prepared the manuscript. All authors reviewed the manuscript.

We declare that we have no competing interests.

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
