## [Reviewer comments · Microbiology Spectrum]

Microbiology Spectrum

Diverse sensory repertoire of paralogous chemoreceptors Tlp2, Tlp3 and Tlp4 in *Campylobacter jejuni*

Taha T, Bassam Elgamoudi, Ekaterina Andrianova, Thomas Haselhorst, Christopher Day, Lauren Hartley-Tassell, Rebecca King, Tahria Najnin, Igor Zhulin, and Victoria Korolik

Corresponding Author(s): Victoria Korolik, Griffith University

Review Timeline:

Submission Date:	September 8, 2022
Editorial Decision:	September 23, 2022
Revision Received:	October 18, 2022
Accepted:	October 19, 2022

Editor: Beile Gao

Reviewer(s): The reviewers have opted to remain anonymous.

Transaction Report:

DOI: <https://doi.org/10.1128/spectrum.03646-22>

September 23, 2022

Prof. Victoria Korolik
Griffith University
Institute for Glycomics
Griffith University, Gold Coast
Southport, Queensland 4222
Australia

Re: Spectrum03646-22 (Diverse sensory repertoire of paralogous chemoreceptors Tlp2, Tlp3 and Tlp4 in *Campylobacter jejuni*)

Dear Prof. Victoria Korolik:

I served as a reviewer for a previous version of this manuscript when it was submitted to *Molecular Microbiology* in March, 2022, and I am an expert on chemotaxis of *C. jejuni*. Basically, I reviewed this manuscript positively for the earlier version and I reviewed carefully for the current version again. Overall, I think this manuscript fits the publication standards of *Microbiology Spectrum*, thus I recommend acceptance after minor modifications.

Taha et al. studied the ligand binding repertoire of 3 paralogous chemoreceptors Tlp2/3/4 in *C. jejuni*. By phylogenetic and comparative genomics analyses, they first demonstrate that Tlp2/3/4 are gene duplication products, with Tlp2 and Tlp4 sharing the same ancestor, and later Tlp3 from Tlp2 duplication. By glycan/small molecule arrays coupled with SPR, they identified and confirmed the binding of multiple amino acids, organic acids, monosaccharides and complex glycans to each Tlp. Then, they examined the binding sites of dCache_1 domain in each Tlp by molecular docking, STD NMR, SPR for ligand competition and point mutations. Single and combinational KO mutants were also generated for Tlp2/3/4 to test the chemotactic motility of *C. jejuni*. The authors concluded that Tlp2/3/4 have overlapping specificities, and likely work cooperatively to balance response to attractants and repellents. Besides, Tlp3-LBD has one major binding pocket with two overlapping but distinctive binding sites for multiple ligands. Overall, this study provides detailed information regarding the evolution, sensory repertoire and molecular mechanism of Tlp2/3/4, also shedding light into functional divergence route of duplicated chemoreceptors.

The following are minor changes for the authors to make. If the authors can correct/improve these following issues, I think the manuscript is suitable for publication in *Microbiology Spectrum*.

Minor Changes:

1. Figure 1 is confusing in terms of the MCP class assignments of Tlp1. Why Tlp1 "homologs" have different H classes? How a Tlp1 with 40H MA domain became 36H or 28H? I think this point is confusing for most readers of your paper. Either explain this in the text or simplify the figure to show the main points - species distribution of Tlp2,3,4 and they all belong to 28H.
2. For Figures 2, 3, 5, 6, 7 with multi-panels, make the ABCD labels uniform.
3. Figures 6 and 7: show statistical analysis.
4. Fig. S2, S3 and S4 show phylogenetic trees based on alignments of full length MCP, sensory domains, and signaling (MA) domains. In Figure S4: Since the MA domains of Tlp2,3,4 are identical, why these MA domains in green/blue/purple colors are in different clusters mixed with Tlp1/11, rather than forming one cluster? Figure S4 legend: "on dCache_1 domain phylogenetic tree (Fig S4).", I think you mean (Fig S3) here.
5. Line 138: Tlp2,3,4-like cluster possessing the amino acid recognition AA motif is shown in Figure S3 not S4 according to your dataset.
6. Line 180: Add a punctuation after the sentence "...sites (10.31)".
7. Line 384: Please make the correction "(Table S6).)."
8. Line 411: Please make the correction "(as shown in Figure ??)."

Thank you for submitting your manuscript to *Microbiology Spectrum*. When submitting the revised version of your paper, please provide (1) point-by-point responses to the issues raised by the reviewers as file type "Response to Reviewers," not in your cover letter, and (2) a PDF file that indicates the changes from the original submission (by highlighting or underlining the changes) as file type "Marked Up Manuscript - For Review Only". Please use this link to submit your revised manuscript - we strongly recommend that you submit your paper within the next 60 days or reach out to me. Detailed instructions on submitting your revised paper are below.

Link Not Available

Below you will find instructions from the *Microbiology Spectrum* editorial office and comments generated during the review.

ASM policy requires that data be available to the public upon online posting of the article, so please verify all links to sequence

records, if present, and make sure that each number retrieves the full record of the data. If a new accession number is not linked or a link is broken, provide production staff with the correct URL for the record. If the accession numbers for new data are not publicly accessible before the expected online posting of the article, publication of your article may be delayed; please contact the ASM production staff immediately with the expected release date.

Sincerely,

Beile Gao

Journals Department
Reviewer comments:

Staff Comments:

Preparing Revision Guidelines

Please return the manuscript within 60 days; if you cannot complete the modification within this time period, please contact me. If you do not wish to modify the manuscript and prefer to submit it to another journal, please notify me of your decision immediately so that the manuscript may be formally withdrawn from consideration by Microbiology Spectrum.

Diverse sensory repertoire of paralogous chemoreceptors Tlp2, Tlp3 and Tlp4 in *Campylobacter jejuni*

Taha, Bassam A. Elgamoudi, Ekaterina P. Andrianova, Thomas Haselhorst, Christopher J. Day, Lauren E. Hartley-Tassell, Rebecca M. King, Tahria Najnin, Igor B. Zhulin & Victoria Korolik

Response to Reviewer's comments:

Reviewer comment 1. Figure 1 is confusing in terms of the MCP class assignments of Tlp1. Why Tlp1 "homologs" have different H classes? How a Tlp1 with 40H MA domain became 36H or 28H? I think this point is confusing for most readers of your paper. Either explain this in the text or simplify the figure to show the main points - species distribution of Tlp2,3,4 and they all belong to 28H.

Author's response: As stated on lines 96-97 "Considering different rates of evolutionary changes observed between sensory and signalling domains, we inferred separate phylogenetic reconstructions for sensory and signaling parts of the chemoreceptors (to make this statement stronger we now cite two publications where, these differences were first described). Fig.1 represents phyletic distribution of dCache_1 domain based on the clustering of this sensory domain on phylogenetic tree shown on Fig S3. Even though some chemoreceptors from Tlp1 group have different length of MCP domain, the respective dCache_1 domain clustered with dCache_1 domain from Tlp1 of *C. jejuni* 11168 (Fig. S3). In these cases, we put them into Tlp1-like group, hypothesizing that these chemoreceptors could bind the same ligands as Tlp1 from *C. jejuni* 11168 regardless of the MCP signaling class they belong to. Differences in the MCP signaling class of Tlp1 might be due to either domain swap or insertion/deletion events that are common for this type of signaling proteins (e.g. ref. 23). To clarify the point, we have added a note on that to the figure legend as well.

Reviewer comment 2. For Figures 2, 3, 5, 6, 7 with multi-panels, make the ABCD labels uniform.

Author's response: Done.

Reviewer comment 3. Figures 6 and 7: show statistical analysis.

Author's response: Figure 6 shows a conventional presentation of competition SPR analysis, typical for sensograms, where the data presented as mean \pm SD (as previously described in ref. 10) and is compared to a theoretical value. Statistical analysis is not meaningful for this type of analysis. We have provided a detailed description of this assay in (Line 179-209 and Line 413-420).

The statistical analysis for figure 7 is now shown.

Reviewer comment 4. Fig. S2, S3 and S4 show phylogenetic trees based on alignments of full length MCP, sensory domains, and signaling (MA) domains. In Figure S4: Since the MA

domains of Tlp2,3,4 are identical, why these MA domains in green/blue/purple colors are in different clusters mixed with Tlp1/11, rather than forming one cluster?

Author's response: Based on the phylogenetic inference and sequence similarity signalling domains evolved at a much slower rates than the sensory domains. We hypothesized (lines 98-99, 116-122), that the reason for this is an adaptation of the signalling domain to other chemosensory components of each taxonomic lineage. Therefore, signalling domains from different receptors that belong to one species are more similar to each other (or even identical), rather than to the same type of receptor in different organisms.

Reviewer comment 5. Figure S4 legend: "on dCache_1 domain phylogenetic tree (Fig S4).", I think you mean (Fig S3) here.

Author's response: Yes, this is correct, apologies for this inadvertent error. It's now been corrected.

Reviewer comment 5. Line 138: Tlp2,3,4-like cluster possessing the amino acid recognition AA motif is shown in Figure S3 not S4 according to your dataset.

Author's response: Yes, this is the same mistake, now corrected.

Reviewer comment 6. Line 180: Add a punctuation after the sentence "...sites (10.31)".

Author's response: We have corrected the error.

Reviewer comment 7. Line 384: Please make the correction "(Table S6).)."

Author's response: We have corrected the error.

Reviewer comment 8. Line 411: Please make the correction "(as shown in Figure ??)."

Author's response: We have corrected the error.

We thank the Reviewer for their helpful comments and attention to detail, allowing us to prepare the best version of this manuscript we can

Victoria Korolik.

October 19, 2022

Prof. Victoria Korolik
Griffith University
Institute for Glycomics
Griffith University, Gold Coast
Southport, Queensland 4222
Australia

Re: Spectrum03646-22R1 (Diverse sensory repertoire of paralogous chemoreceptors Tlp2, Tlp3 and Tlp4 in *Campylobacter jejuni*)

Dear Prof. Victoria Korolik:

Your manuscript has been accepted, and I am forwarding it to the ASM Journals Department for publication. You will be notified when your proofs are ready to be viewed.

Sincerely,

Beile Gao
Editor, Microbiology Spectrum

Journals Department
Supplemental Material: Accept
Supplemental Movie: Accept
Supplemental Movie: Accept